# LC–MS/MS Analysis of Choline Compounds in Japanese-Cultivated Vegetables and Fruits

**DOI:** 10.3390/foods9081029

**Published:** 2020-07-31

**Authors:** Wenhao Wang, Shohei Yamaguchi, Masahiro Koyama, Su Tian, Aya Ino, Koji Miyatake, Kozo Nakamura

**Affiliations:** 1Department of Science and Technology, Graduate School of Medicine, Science and Technology, Shinshu University, 8304, Minamiminowa, Nagano 399-4598, Japan; 20hs502e@shinshu-u.ac.jp (W.W.); 19hs505d@shinshu-u.ac.jp (S.Y.); 2Wellnas Co., Ltd., Toranomon Masters Building 6F, 1-12-14, Toranomon, Minato-ku, Tokyo 105-0001, Japan; mkoyama32@wellnas.biz; 3Department of Nutrition and Food Hygiene, School of Public Health, Hebei Medical University, Shijiazhuang 050017, China; sutianjia@yahoo.co.jp; 4Kochi Agricultural Research Center, 1100 Hataeda, Nankoku, Kochi 783-0023, Japan; aya_ino@ken4.pref.kochi.lg.jp; 5Institute of Vegetable and Floriculture Science, NARO, 360 Kusawa, Ano-cho, Tsu, Mie 514-2392, Japan; miya0424@affrc.go.jp; 6Institute of Agriculture, Academic Assembly, Shinshu University, 8304, Minamiminowa, Nagano 399-4598, Japan

**Keywords:** choline esters, acetylcholine, propionylcholine, butyrylcholine, choline, LC–MS/MS, eggplant

## Abstract

Choline is an essential nutrient and choline esters are potential functional food ingredients. We aimed to analyze the choline compound content in 19 cultivated fruits and vegetables and identify those with high acetylcholine content. We utilized liquid chromatography with tandem mass spectrometry to quantify choline compounds according to the standard addition method. Choline compounds were extracted from lyophilized fruit/vegetable powders and passed through a weakly acidic cation exchange column, resulting in a concentrated solution of choline compounds. The compounds were separated on a pentafluorophenyl column and then analyzed using positive mode electrospray ionization. Results showed that acetylcholine and choline were the primary choline compounds in all agricultural products; propionylcholine and butyrylcholine were minor compounds in 17 and 12 agricultural products, respectively. The acetylcholine concentration was 2900-fold higher in eggplants (6.12 mg/100 g fresh weight [FW]) than in other agricultural products (average: 2.11 × 10^−3^ mg/100 g FW). The concentration of acetylcholine differed only 2-fold between eggplant cultivars with the highest (′Higomurasaki′: 5.53 mg/100 g FW) and lowest (′Onaga nasu′: 2.79 mg/100 g FW) concentrations. The half-life of acetylcholine in eggplants was approximately 16 days, which is longer the shelf life of eggplants. Thus, eggplants can be a good source of acetylcholine.

## 1. Introduction

Choline, a functional compound originally isolated from pig and ox bile by Adolph Strecker [1], is an essential nutrient. It has been reported that rats fed with choline-deficient foods have fatty livers and develop cirrhosis; and in humans, the long-term central vein administration of choline-free parenteral nutrition causes liver fat deposits that lead to hepatocyte damage [2,3]. Furthermore, choline deficiency can cause loss of mitochondrial membrane potential [4], which can result in apoptosis [5,6].

Choline esters are choline derivatives that may also be functional food ingredients. Lactoylcholine (LCh) has been reported in lactic acid bacteria-fermented foods; it has an excellent antihypertensive effect on spontaneously hypertensive rats [7,8]. Similarly, we recently reported the antihypertensive effect of acetylcholine (ACh)-containing eggplants [9].

ACh (Figure 1) is a widely known choline ester that is also a neurotransmitter. It exists not only in the nervous systems of mammals, but also in those of many other organisms [10,11]. Ewins [12] first discovered it as a natural blood pressure-lowering compound through intravenous injections of ergot. Loewi [13] then found that ACh is a chemical transmitter that affects neural activity. To that end, Dale [14] showed that ACh is released from parasympathetic terminals, and is a neurotransmitter.

The above information suggests that choline compounds, including choline and choline esters, are widely present in nature and may contribute to food functions. According to the United States Department of Agriculture Food Composition Databases [15], more than 700 cultivated vegetables and fruits and their processed products contain choline. Zeisel et al. [16] investigated the choline content of 145 common foods and found that dried soybeans have the highest content (116 mg/100 g). In contrast, the choline ester content in cultivated vegetables and fruits has rarely been studied. In our survey, only one study was found that reported the levels of ACh in eggplants [10], and it remains unclear whether ACh concentration in eggplants is high or low relative to the concentrations found in other cultivated products. To determine whether the content of choline compounds in eggplant has advantages over other cultivated vegetables and fruits, in this study, we investigated choline compounds (i.e., choline, ACh, propionylcholine [PCh] and butyrylcholine [BCh]) in 19 types of Japanese-cultivated vegetables and fruits, and compared choline compound contents to identify cultivated crops with the highest levels of ACh.

## 2. Materials and Methods

### 2.1. Chemicals

Ultrapure water with a specific resistance of 18.2 MΩ/cm was produced in an Arium 611 Ultrapure Water System (Sartorius Co., Goettingen, Germany) and used in our experiments. Methanol (high-performance liquid chromatography [HPLC]-grade), acetonitrile (HPLC-grade), choline chloride, propionic acid, n-butyric acid, ethyl acetate (EtOAc), formic acid and 1-N hydrochloric acid were purchased from Nacalai Tesque, Inc. (Kyoto, Japan). *N*,*N*′-dicyclohexylcarbodiimide (DCC) was purchased from Peptide Institute, Inc. (Osaka, Japan); 4-N HCl/dioxane and *N*,*N*-dimethylformamide (DMF) were purchased from Watanabe Chemical Industries, Ltd. (Hiroshima, Japan); *N*,*N*-dimethyl-4-aminopyridine (DMAP) was purchased from Kokusan Chemical Co., Ltd. (Tokyo, Japan); and ACh chloride, pivalic acid and deuterium oxide were purchased from Kanto Chemical Co., Inc. (Tokyo, Japan). (2-aminoethyl)trimethylammonium chloride hydrochloride was purchased from Sigma-Aldrich, Inc. (Ontario, Canada). All other purchased chemicals—including dichloromethane (DCM), sodium dihydrogen phosphate and disodium hydrogen phosphate—were from Fujifilm Wako Pure Chemical Industries, Ltd. (Osaka, Japan). PCh, BCh, LCh and (2-aminoethyl)trimethylammonium pivaloylamide (EN) were synthesized in our laboratory.

### 2.2. General Techniques

Liquid chromatography–tandem mass spectrometry (LC–MS/MS), nuclear magnetic resonance (NMR) and matrix-assisted laser desorption/ionization time of flight mass spectrometry (Maldi-TOF MS) analyses were performed at Research Center for Supports to Advanced Science, Shinshu University. LC–MS/MS analysis of choline compounds were performed on a Quattro micro API (MS) with an Acquity UPLC system (Waters, Co., USA). NMR spectra were recorded using a Bruker DRX 500 spectrometer (Bruker BioSpin Corp., Billerica, MA, USA) at 500 MHz for ^1^H and 126 MHz for ^13^C NMR at 25 °C. High-resolution Maldi-TOF MS analysis was performed on an AB Sciex TOF/TOF 5800 equipped with a 1 kHz neodymium: yttrium–aluminum–garnet laser (AB Sciex, Framingham, MA, USA).

### 2.3. Synthesis of PCh, BCh, LCh and EN

LCh was synthesized following the protocol in our previous report [8]. PCh, BCh and EN were synthesized following the methods below.

#### 2.3.1. Synthesis of PCh

Choline chloride (molecular weight [MW] = 139.62, 140 mg, 1.00 mmol) was dissolved in DMF (2.00 mL) by dropwise addition of propionic acid (2.00 eq, MW = 74.08, density = 0.990 g/cm^3^, 150 µL, 2.00 mmol), after which DCC (2.00 eq, MW = 206.33, 413 mg, 2.00 mmol) and DMAP (0.100 eq, MW = 122.17, 12.0 mg, 0.100 mmol) were added; the mixture was stirred at 25 °C ± 5 °C for 2 h. Then, 4-N HCl/dioxane (0.250 mL) was added, and *N*,*N*′-dicyclohexylurea was filtered out with a suction filter. Afterward, the filtrate was concentrated under reduced pressure. EtOAc (10.0 mL) was added to induce crystallization, which was followed by the removal of the supernatant by decantation. The PCh obtained from recrystallization (DCM:EtOAc ratio of 1:1) appeared as white crystals, with a yield of 92.0% (0.920 mmol, 180 mg, hydrochloride salt).

#### 2.3.2. Synthesis of BCh

Choline chloride (MW = 139.62, 140 mg, 1.00 mmol) was dissolved in DMF (2.00 mL) by dropwise addition of n-butyric acid (2.00 eq, MW = 88.11, density = 0.960 g/cm^3^, 184 µL, 2.00 mmol), after which DCC (2.00 eq, MW = 206.33, 413 mg, 2.00 mmol) and DMAP (0.100 eq, MW = 122.17, 12.0 mg, 0.100 mmol) were added; the mixture was stirred at 25 °C ± 5 °C for 2 h. Then, 4-N HCl/dioxane (0.250 mL) was added, and *N*,*N*′-dicyclohexylurea was filtered off with a suction filter. Afterward, the filtrate was concentrated under reduced pressure. EtOAc (10.0 mL) was added to induce crystallization, which was followed by the removal of the supernatant by decantation. The BCh obtained by recrystallization (DCM:EtOAc ratio of 1:1) appeared as white crystals, with a yield of 89.1% (0.891 mmol, 187 mg, hydrochloride salt).

#### 2.3.3. Synthesis of EN

(2-aminoethyl)trimethylammonium chloride hydrochloride (1.00 eq, MW = 175.10, 175.10 mg, 1.00 mmol) was dissolved in acetonitrile (2.00 mL) by dropwise addition of pivalic acid (2.00 eq, MW = 102.13, 204.26 mg, 2.00 mmol), after which DCC (2.00 eq, MW = 206.33, 412.66 mg, 2.00 mmol) and DMAP (0.100 eq, MW = 122.17, 12.22 mg, 0.100 mmol) were added; the mixture was stirred at 25 °C ± 5 °C for 4 h. Then, the solution was suction filtered, and the filtrate was concentrated under reduced pressure. EtOAc (10.0 mL) was added to induce crystallization, which was followed by the removal of the supernatant by decantation. The EN obtained by recrystallization (DCM:EtOAc ratio of 1:1) appeared as white crystals, with a yield of 96.1% (0.961 mmol, 214 mg, hydrochloride salt). The spectral databases of PCh, BCh and EN are described in Table 1.

### 2.4. Samples of Fresh Cultivated Vegetables and Fruits

The samples were all purchased in the market, their sources were as follows: ‘Zubari 163’ cucumbers, ‘Rinka 409’ tomatoes, ‘Special’ paprika, ‘Bell-masari’ bell peppers, ‘Senryo No. 2’ eggplants, ‘Welcome’ asparagus, ‘Nagaimo’ Japanese yam, ‘Shinshu 868’ cabbage, ‘Shinano hope’ lettuce and ‘Kouyou No. 2’ carrots were from Shiojiri City (Nagano Prefecture, Japan); ‘Manganji togarashi’ Shishito peppers and ‘Twentieth century’ Japanese pears were from Ina City (Nagano Prefecture, Japan); ‘Shinano dolce’ apples were from Matsumoto City (Nagano Prefecture, Japan); ‘Nagano purple’ grapes were from Minamiminowa Village (Nagano Prefecture, Japan); kaiware daikon, broccoli sprouts, alfalfa bean sprouts, pea sprout and buckwheat sprouts were from Salada Cosmo Co., Ltd. (Japan); ‘Senshu mizunasu’ eggplants were from Izumi City (Osaka Prefecture, Japan); ‘Batten nasu’ eggplants were from Uki City (Kumamoto Prefecture, Japan); ‘Koryo sarada nasu’ eggplants were from Kitaibaraki-gun (Nara Prefecture, Japan); ‘Onaga nasu’ eggplants were from Yamaga City (Kumamoto Prefecture, Japan); ‘Chikuyo’ eggplants were from Kumamoto City (Kumamoto Prefecture, Japan); and ‘Higomurasaki’ eggplants were from Aso-gun (Kumamoto Prefecture, Japan).

### 2.5. Sample Preparation for Quantification

After the roots, seeds, peel and calyxes were removed, edible portions of the fruits and vegetables were sliced into pieces, 1–3 cm in width. The sections were stored at −80 °C and then lyophilized in a freeze dryer (FDU-2000; Tokyo Rikakikai Co., Ltd., Tokyo, Japan). The dry yields are listed in Table 2.

The lyophilized products were ground in a mill mixer (MNN-2001; Tokyo Unicom, Tokyo, Japan) for 1.5 min (31,000 rpm) to prepare lyophilized powders for sample extraction. EN (10.0 μL, internal standard) and 10.0-mM phosphate-buffered saline (PBS, 190 μL, 3.40-mmol/L sodium dihydrogen phosphate, and 6.60-mmol/L disodium hydrogen phosphate) were added to the lyophilized powder (10.0 mg) and vortexed for 3 min; the supernatant was recovered after centrifugation (1000× *g*, room temperature, 3 min). Then, 10.0-mM PBS (200 μL) was added to the residue and vortexed; the supernatant was recovered after centrifugation. This extraction process was repeated twice, and the collected supernatants were combined (approximately 600 μL total).

Next, solid-phase extraction was performed on the extracted samples with a weakly acidic cation exchange cartridge (InertSep CBA 100 mg/1 mL; GL Sciences, Tokyo, Japan). The exchange cartridge was activated with methanol (1.00 mL) and then equilibrated with 10.0-mM PBS. The extracted samples were then added to the cartridge, and the choline compounds were adsorbed using 10.0-mM PBS (600 μL). After rinsing with water to remove contaminants, we eluted the adsorbed choline compounds with 1-N hydrochloric acid (500 μL). The eluate was placed in a 1.00-mL volumetric flask and then diluted to the volume mark with the mobile phase (33.0% *v*/*v* methanol containing 0.0100% formic acid). The resulting solution was divided into three portions (A, B, C, 300 μL each), and LC–MS/MS was used to confirm the area of sample A, then standard solutions of different concentrations were added to samples B and C (B = 1.5 × A; C = 2 × A), creating three solutions with different concentrations of choline compounds to be used for quantification by the standard addition method (because choline was compared with choline ester, the sample needed to be diluted ten times when quantifying choline amount).

Six fruits from a ‘Higomurasaki’ eggplant were used to investigate the half-life of ACh in eggplants. Each fruit was packaged in wet paper and stored at 25 °C ± 5 °C. A portion of each fruit was removed daily, from day 1 to 5, for ACh quantification (*n* = 3).

### 2.6. Quantification of Choline Compounds

Choline compound content of fresh crops was analyzed for following a previously published method [9], which utilized LC–MS/MS. The separation was achieved by using 33.0% (*v*/*v*) methanol containing 0.0100% formic acid as the mobile phase at flow rates of 0.5 mL/min (LC) and 0.3 mL/min (MS). The column was a YMC-Triart PFP (4.6 mm × 250 mm, 5 μm) set at 40 °C. The injection volume was 50.0 μL, and the analysis time was 30 min. The mass spectrometer was operated in the positive mode with electrospray ionization. The capillary voltage was 3.500 V, cone voltage was 10 V, collision voltage was 10 V, N₂ gas flow (desolvation) was 600 L/h, N₂ gas flow (cone) was 50 L/h, N₂ source temperature was 120 °C, and N₂ desolvation temperature was 350 °C. The multiple reaction monitoring mode transitions in the mass-to-charge ratio (*m*/*z*) for each choline compound were as follows: 187.18 → 128.15 (EN); 146.10 → 87.0 (ACh); 160.10 → 101.0 (PCh); 174.10 → 115.0 (BCh); 176.10 → 117.05 (LCh); and 104.20 → 60.2 (choline).

A calibration curve was prepared using the peak area values obtained by LC–MS/MS analysis, and the choline compounds were quantified using the standard addition method. The concentration of each choline compound was corrected based on the recovery of EN (internal standard substances). We used the obtained concentrations to calculate the choline compound content of lyophilized powders (mg/g dry weight [DW]), and the amount of each choline compound per 100 g of fresh weight (µg/100 g FW) was calculated from the yield.

### 2.7. Method Validation

The validity of our method was evaluated using linearity (coefficient of determination; *R*^2^), limit of detection (LOD), limit of quantification (LOQ), precision (intra- and inter-day reproducibility) and accuracy (recovery and relative standard deviation). The details of our validation method are described as in the following sections.

#### 2.7.1. Linearity

Standard solutions of the choline esters (ACh, PCh, BCh and LCh) were prepared and adjusted to the following concentrations: 0.0100, 0.125, 0.250, 0.500, 1.00 and 3.00 μg/mL. Because of the high content of choline in food, concentrations of the choline standard solution were adjusted to 0.0100, 0.125, 0.250, 0.500, 1.00, 3.00, 15.0 and 30.0 μg/mL. The concentrations of the solutions were adjusted by mixing each choline compound standard with the mobile phase. The standard solutions were analyzed using LC–MS/MS. A regression line was created from the resulting peak area values and the concentrations of each standard choline compound solution. Then, *R*^2^ was calculated to evaluate linearity.

#### 2.7.2. LOD and LOQ

Standard solution: We prepared 0.250 μg/mL of each standard choline compound solution (*n* = 3) and analyzed them following the recommended formulas by the International Union of Pure and Applied Chemistry (IUPAC Compendium of Chemical Terminology Gold Book) [17] to calculate LOD and LOQ as follows: LOD = 5.84 σ/S and LOQ = 20 σ/S; where σ is the standard deviation, and S is the slope of the calibration curve.

Eggplant samples: We used the same eggplant sample three times in a row for quantification and calculated the LOD and LOQ in the same way as the standard solution.

#### 2.7.3. Precision

Standard solution: Precision and accuracy were evaluated by mixing 0.500 μg/mL of each standard choline compound (ACh, PCh, BCh, LCh and choline) with 10.0-mM PBS. We evaluated the precision of the method by analyzing the percent relative standard deviation (RSD%) for intraday and inter-day reproducibility. Intraday reproducibility was assessed by the RSD% of the same-day quantitative values for each standard 0.500-μg/mL choline compound solution (*n* = 3). Interday reproducibility was assessed by calculating the RSD% of the three days of quantitative values for every 0.500-μg/mL standard choline compound solution (quantitative values recorded daily; *n* = 3 each day).

Eggplant samples: Eggplant samples were used continuously for three days for quantification, three times each day. The intraday reproducibility was evaluated by the RSD% of the daily quantitative value (*n* = 3). Interday reproducibility was assessed by the RSD% of the three-day quantitative value (quantitative values recorded daily; *n* = 3 each day).

#### 2.7.4. Accuracy

Standard solution: Accuracy was determined using LC–MS/MS to analyze every 0.500 μg/mL of the standard choline compound solution (*n* = 5). The average value of the quantitative concentration was divided by the theoretical concentration to obtain the recovery. In addition, RSD% was calculated for the quantitative values (*n* = 5).

Eggplant samples: The lyophilized eggplant powder was divided into two parts, one of which was added to 102 μL of the standard solution (ACh: 2.00 μg, Ch: 30.0 μg, PCh: 0.00180 μg, BCh: 0.00180 μg; *n* = 3) and the other was not (*n* = 3). The sample concentration without the added standard solution was subtracted from the sample concentration with the added standard solution, and the result was then divided by the theoretical added concentration to obtain the recovery rate (*n* = 3). The average value was taken, and RSD% was calculated.

### 2.8. Statistical Analysis

All results were expressed as the mean ± standard error. Analysis of variance revealed that the *p*-values for all results were <0.05, which indicates statistical significance. Standard differences were considered significant at *p* < 0.05 when evaluated by the Tukey’s honestly significant difference test and Student′s *t*-test.

## 3. Results

### 3.1. Quantification of Choline Compounds in Fresh Cultivated Vegetables and Fruits

The results of method validation are shown in Table 3. The results refer to the standards of the US Food and Drug Administration guidelines [18]. We ensured that the *R*^2^ of the calibration curve obtained by the standard addition method was greater than 0.99 each time and the RSD% of all samples was less than 15.0%.

Quantification revealed that all vegetables and fruits analyzed in this study contained ACh and choline (Table 4), but LCh was not detected. This was because LCh is usually produced by lactic acid bacteria, so it does not exist in fresh crops [8]. PCh was found in 17 types of fresh crops, but not in tomatoes and pea sprouts. BCh was present in 12 crops. The most abundant choline compound was choline. The most abundant choline ester was ACh (average of 0.324 mg/100 g FW). Minor choline esters included PCh and BCh, which were present in 17 and 12 types of agricultural products; the percentages of each minor choline ester to all choline esters was 2.19% and 0.623%, respectively.

Eggplants had the highest concentration of choline esters (6.12 mg/100 g FW), followed by Japanese yams (8.43 × 10^−2^ mg/100 g FW) and Shishito pepper (5.47 × 10^−3^ mg/100 g FW). As shown in Table 4, the ACh and PCh content of eggplants significantly differed from those of other fresh crops (*p* < 0.01) and the BCh content of eggplants significantly differed from that of seven varieties of cultivated vegetables (cucumbers, bell peppers, Shishito peppers, Japanese yams, kaiware daikon, broccoli sprouts and buckwheat sprouts; *p* < 0.01). In addition, the choline content of eggplants significantly differed from that of Shishito peppers (*p* < 0.05) and other fresh cultivated vegetables (*p* < 0.01), except for cucumbers, paprika and Japanese pears.

It should be noted that the ACh content of eggplants was 2900-fold higher than the average ACh content of other fresh crops (2.11 × 10^−3^ mg/100 g FW) and more than 3000-fold the ACh content of other solanaceous plants (tomatoes, paprika, bell peppers and Shishito peppers; average: 2.53 × 10^−3^ mg/100 g FW).

Next, because eggplant contains extremely high levels of ACh and to confirm whether this was a unique situation for Senryo No. 2, we randomly selected six eggplant varieties from different Japanese regions to study the difference in their choline compound content.

### 3.2. Quantification of Choline Compounds in Six Cultivars of Eggplants

A quantitative analysis of the choline compounds in six different Japanese cultivars of eggplants available on the market—’Higomurasaki’, ‘Chikuyo’, ‘Senshu mizunasu’, ‘Koryo sarada nasu’, ‘Batten nasu’ and ‘Onaga nasu’—was performed using LC–MS/MS (*n* = 3, Table 5). ACh, PCh, BCh and choline were present in all six cultivars. Among the cultivars, ‘Higomurasaki’ had the highest levels of choline esters (ACh: 5.53 × 10^3^ µg/100 g FW; PCh: 7.78 µg/100 g FW; BCh: 2.85 µg/100 g FW) and ‘Batten nasu’ had the highest levels of choline (3.63 × 10^4^ µg/100 g FW). The difference in the ACh content between cultivars with the highest and the lowest ACh levels was merely 2-fold (Table 5), suggesting that all eggplant cultivars are richer in ACh than other fresh cultivated vegetables.

### 3.3. Effect of Storage on Choline Compound Content in Eggplants (′Higomurasaki′)

LC–MS/MS was used to investigate the effect of storage on the ACh content of ‘Higomurasaki’ eggplants (*n* = 3; Figure 2), which had the highest ACh levels in this study. Results showed that ACh levels gradually declined from the 1st day (5.53 ± 0.17 mg/100 g FW) to the 5th day (4.48 ± 0.05 mg/100 g FW). The difference in ACh concentrations between the 1st and 5th days was significant. Therefore, we calculated that the half-life of ACh in eggplants was approximately 16 days. Generally, when eggplant fruits are stored at approximately 25 °C, the fruits stay fresh for 2–3 days [19].

## 4. Discussion

In this study, we used LC–MS/MS to determine the concentration of choline compounds in 19 types of fresh fruits and vegetables. This is the first report comparing the levels of ACh in fruits and vegetables that are ingested daily. Results showed the existence of choline compounds in all agricultural products. of those compounds, the most abundant was choline, which was present in all agricultural products and was a common nutritional compound.

Choline esters were also widely found. The most abundant choline ester was ACh. In that regard, our results concurred with those of a previous study, in which the ACh content of eggplants was 417 nmol/g (6.10 mg/100 g FW) [10] when quantified by HPLC–electrochemical detection. Like choline, ACh was present in all the agricultural products, a fact that had not previously been reported. Further analyses revealed that the concentration of ACh was 2900-fold greater in eggplants than in 18 other agricultural products. This study was the first to show the magnitude of the differences in the ACh content between eggplants and other crops, which is significant because, in our 2019 clinical study, we found that continuous intake of eggplant powder containing 2.3 mg of eggplant-derived choline ester (acetylcholine) can improve blood pressure and psychological conditions [20]. In this study, eggplants 1) did not contain LCh, 2) had low PCh and BCh levels and 3) were rich in ACh. A small piece of eggplant have the potential to lower blood pressure; however, it would be difficult to achieve that effect with other plants because of their comparatively low ACh content. To our knowledge, this is the first study to show the existence of the minor choline esters PCh and BCh in agricultural products. Moreover, we found that PCh and BCh were widely present in plants.

Eggplant (*Solanum melongena* L., family Solanaceae), an agronomically and economically important crop that has been planted in South Asia and East Asia since prehistory, is believed to have originated from eastern India [21]. Today, about 50 million tons of cultivated eggplant is grown on more than 1.800.000 ha worldwide [22]. Eggplant is recognized as a high-fiber, non-starchy vegetable [23,24,25]. Therefore, it has been the focus of clinical cholesterol reduction experiments in patients with hypercholesterolemia [26] and body fat reduction experiments in women who are overweight [27]. In addition to ACh, eggplants contain other functional components, such as nasunin [28], γ-aminobutyric acid [29] and chlorogenic acid [30]. Eggplants reportedly have cardioprotective [31] and antioxidant [32] effects in rats.

In conclusion, choline compounds are widely available in agricultural products, which suggests that fresh fruits and vegetables can be consumed daily as a source of such compounds. The levels of choline and ACh are generally high in eggplants, and the half-life of eggplants is longer than their shelf life, which proves the feasibility of using eggplants as functional foods. The decrease in ACh during storage may be because ACh is gradually degraded by the ACh degrading enzyme (cholinesterase) originally contained in eggplants. Our results enable the use of eggplants as an effective source of ACh. However, owing to the unstable chemical nature of ACh and its high hydrophilicity, it can be lost during washing, boiling or other cooking processes, resulting in variations in the calculated intake. Therefore, we recommend that future studies focus on the effects of cooking on ACh content in eggplants.

## Figures and Tables

**Figure 1 foods-09-01029-f001:**
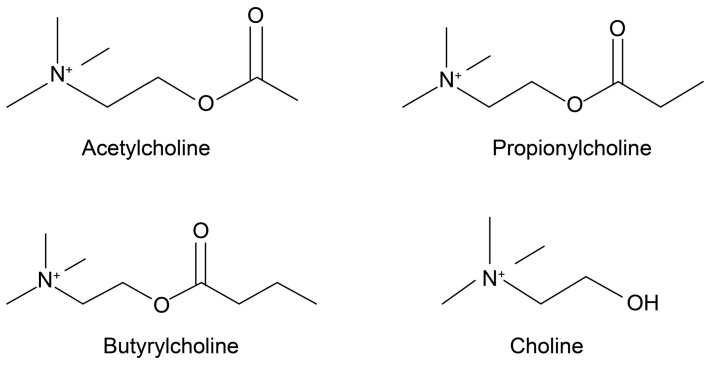
Chemical structures of ACh, PCh, BCh and choline. ACh—acetylcholine; PCh—propionylcholine; BCh—butyrylcholine.

**Figure 2 foods-09-01029-f002:**
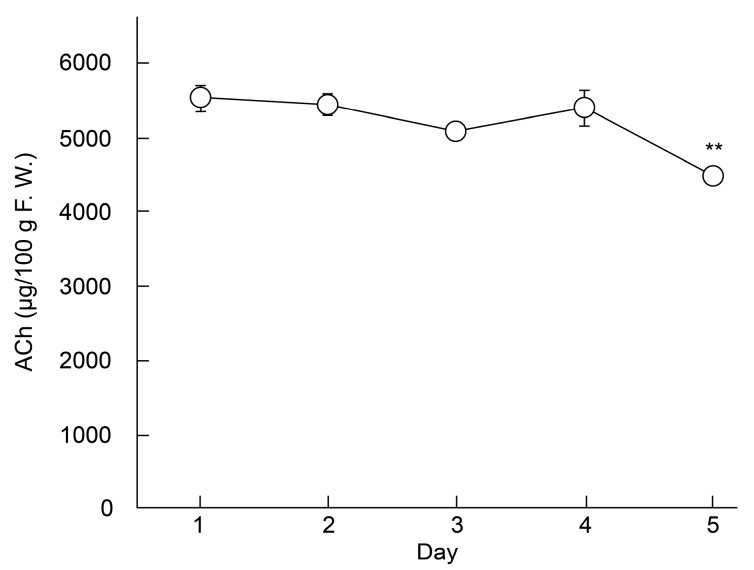
Changes in the ACh content in the fruits of an eggplant (′Higomurasaki′) stored at 25 °C ± 5 °C. Each data point and error bar represent the mean ± standard error. ** *p* < 0.01 versus day 1, evaluated using the Student′s *t*-test. ACh—acetylcholine.

**Table 1 foods-09-01029-t001:** Spectral data for synthesized PCh, BCh, EN.

^1^H, ^13^C NMR in D_2_O (ppm)	PCh					BCh					EN				
	δ_C_	Type	δ_H_	Multi (*J* in Hz)	Type	δ_C_	Type	δ_H_	Multi (*J* in Hz)	Type	δ_C_	Type	δ_H_	Multi (*J* in Hz)	Type
Chemical shift	8.0	CH_3_	0.98	*t* (7.5)	CH_3_	12.7	CH_3_	0.74	*t* (7.5)	CH_3_	26.3	CH_3_	1.04	*s*	CH_3_
	27.1	CH_2_	2.34	*q* (7.5)	CH_2_	17.9	CH_2_	1.45	*sextet* (7.4)	CH_2_	33.6	CH_2_	3.04	*s*	CH_3_
	53.7	CH_3_	3.10	*s*	CH_3_	35.5	CH_2_	2.25	*t* (7.5)	CH_2_	38.4	C	3.34	*m*	CH_2_
	58.2	CH_2_	3.63	*m*	CH_2_	53.7	CH_3_	3.03	*s*	CH_3_	53.2	CH_3_	3.55	*m*	CH_2_
	64.5	CH_2_	4.44	*m*	CH_2_	58.1	CH_2_	3.56	*m*	CH_2_	64.2	CH_2_			
	176.4	C				64.5	CH_2_	4.38	*m*	CH_2_	182.9	C			
						176.4	C								
MALDI–TOF MS	found			required		found			required		found			required	
[M]^+^ (*m*/*z*)	160.1327			160.1332		174.1479			174.1489		187.1809			187.1805	

PCh—propionylcholine; BCh—butyrylcholine; EN—(2-aminoethyl)trimethylammonium pivaloylamide; δ—chemical shift; ^1^H NMR—proton nuclear magnetic resonance; ^13^C NMR—carbon-13 nuclear magnetic resonance.

**Table 2 foods-09-01029-t002:** Fresh weight, dry weight and lyophilized yield of fresh cultivated vegetables and fruits.

Crop	Cultivar	Fresh Weight (g)	Dry Weight (g)	Yield (%)
Eggplant	Senryo No. 2	56.59	4.49	7.93
Eggplant	Senshu mizunasu	89.24	4.73	5.30
Eggplant	Batten nasu	61.21	3.78	6.18
Eggplant	Koryo sarada nasu	41.21	2.40	5.82
Eggplant	Onaga nasu	131.96	7.98	6.05
Eggplant	Chikuyo	64.30	4.13	6.42
Eggplant	Higomurasaki	121.78	6.96	5.72
Cucumber	Zubari 163	114.52	5.90	5.15
Tomato	Rinka 409	95.37	4.99	5.23
Paprika	Special	90.61	8.90	9.82
Bell pepper	Bell-masari	25.96	3.07	11.83
Shishito pepper	Manganji togarashi	22.19	1.99	8.97
Asparagus	Welcome	17.60	1.45	8.24
Japanese yam	Nagaimo	79.76	21.23	26.62
Cabbage	Shinshu 868	129.60	7.64	5.90
Lettuce	Shinano hope	79.19	2.85	3.60
Carrot	Kouyou No. 2	113.68	13.98	12.30
Kaiware daikon	unknown	26.99	1.44	5.34
Broccoli sprout	unknown	21.92	1.12	5.52
Alfalfa bean sprout	unknown	32.17	1.70	5.28
Pea sprout	unknown	33.47	2.63	7.86
Buckwheat sprout	unknown	15.89	1.12	7.05
Apple	Shinano dolce	71.23	10.07	14.14
Japanese pear	Twentieth century	80.59	10.59	13.14
Grape	Nagano purple	71.54	14.34	20.04

yield (%)—dry weight/fresh weight × 100.

**Table 3 foods-09-01029-t003:** Method validation.

Types	Choline Compounds	Linearity		Precision		Accuracy		Limit	
		Range (μg/mL)	*R* ^2^	Intraday (%)	Interday (%)	Recovery (%)	RSD (%)	LOD (pmol/mL)	LOQ (pmol/mL)
Standard solutions	ACh	0.0100–3.00	1.00	0.0727	0.521	96.4	0.471	38.7	133
	PCh	0.0100–0.500	0.999	0.915	1.29	97.3	0.929	70.6	242
	BCh	0.0100–0.500	0.998	0.384	0.676	97.1	1.24	77.0	264
	LCh	0.0100–6.00	0.997	0.756	1.58	96.2	1.10	42.1	144
	Ch	0.0100–30.0	0.999	0.483	1.15	97.4	0.693	53.6	184
Eggplant samples	ACh			0.805	7.28	84.0	6.30	6.73 × 10^3^	2.31 × 10^4^
	PCh			1.80	5.39	95.7	12.4	3.23	11.1
	BCh			3.31	4.18	91.3	11.6	8.63	29.5
	Ch			3.19	14.5	81.4	0.240	2.12 × 10^4^	7.27 × 10^4^

ACh—acetylcholine; PCh—propionylcholine; BCh—butyrylcholine; LCh—lactoylcholine; Ch—choline; RSD—relative standard deviation; LOD—limit of detection; LOQ—limit of quantification.

**Table 4 foods-09-01029-t004:** Choline compound content in fresh cultivated vegetables and fruits (*n* = 3).

Crop	Cultivar	Status	Choline Compounds							
			ACh		PCh		BCh		Ch	
Cucumber	Zubari 163	μg/100 g FW	7.79 × 10^−1^	±0.01 **	10.1	±0.25 **	2.10	±0.05 **	3.24 × 10^4^	±502
		μg/100 g DW	15.1	±0.18 **	196	±4.79 **	40.7	±1.03 **	6.29 × 10^5^	±9751 **
Tomato	Rinka 0409	μg/100 g FW	8.11 × 10^−1^	±0.05 **	ND		ND		7.57 × 10^4^	±2130 **
		μg/100 g DW	15.5	±0.90 **					1.45 × 10^6^	±40,706 **
Paprika	Special	μg/100 g FW	1.80	±0.04 **	4.31 × 10^−1^	±0.01 **	3.47 × 10^−1^	±0.00	3.40 × 10^4^	±88
		μg/100 g DW	18.3	±0.43 **	4.39	±0.10 **	3.53	±0.05	3.46 × 10^5^	±892
Bell pepper	Bell-masari	μg/100 g FW	5.95	±0.16 **	1.20	±0.03 **	3.98	±0.03 **	8.40 × 10^4^	±1543 **
		μg/100 g DW	50.3	±1.33 **	10.1	±0.28 **	33.7	±0.25 **	7.11 × 10^5^	±13,043 **
Shishito pepper	Manganji togarashi	μg/100 g FW	1.55	±0.03 **	2.86	±0.06 **	5.47	±0.24 **	4.31 × 10^4^	±375 *
		μg/100 g DW	17.3	±0.38 **	31.9	±0.72 **	61.0	±2.67 **	4.81 × 10^5^	±4187
Eggplant	Senryo No. 2	μg/100 g FW	6.12 × 10^3^	±132	6.25	±0.09	5.26 × 10^−1^	±0.01	2.91 × 10^4^	±968
		μg/100 g DW	7.71 × 10^4^	±1668	78.8	±1.15 **	6.62	±0.12	3.67 × 10^5^	±12,204
Asparagus	Welcome	μg/100 g FW	2.04	±0.02 **	15.2	±0.27 **	1.90 × 10^−1^	±0.01	9.69 × 10^4^	±1964 **
		μg/100 g DW	24.7	±0.19 **	184	±3.26 **	2.31	±0.07	1.18 × 10^6^	±23,836 **
Japanese yam	Nagaimo	μg/100 g FW	2.87	±0.06 **	84.3	±0.18 **	3.79	±0.11 **	8.18 × 10^4^	±1130 **
		μg/100 g DW	10.8	±0.24 **	317	±0.69 **	14.3	±0.41 *	3.07 × 10^5^	±4245
Cabbage	Shinshu 868	μg/100 g FW	6.83 × 10^−1^	±0.02 **	9.49 × 10^−1^	±0.03 **	3.32 × 10^−1^	±0.01	5.55 × 10^4^	±2908 **
		μg/100 g DW	11.6	±0.39 **	16.1	±0.59 **	5.64	±0.20	9.41 × 10^5^	±49,337 **
Lettuce	Shinano hope	μg/100 g FW	3.32 × 10^−1^	±0.01 **	7.73 × 10^−2^	±0.00 **	ND		4.40 × 10^4^	±790 **
		μg/100 g DW	9.22	±0.29 **	2.15	±0.03 **			1.22 × 10^6^	±21,959 **
Carrot	Kouyou No. 2	μg/100 g FW	2.22	±0.11 **	7.95 × 10^−1^	±0.04 **	6.44 × 10^−1^	±0.04	1.22 × 10^5^	±3068 **
		μg/100 g DW	18.0	±0.86 **	6.47	±0.36 **	5.23	±0.29	9.88 × 10^5^	±24,946 **
Kaiware daikon	unknown	μg/100 g FW	8.64 × 10^−1^	±0.03 **	2.51 × 10^−1^	±0.01 **	3.16	±0.14 **	7.72 × 10^4^	±1441 **
		μg/100 g DW	16.2	±0.52 **	4.70	±0.11 **	59.2	±1.55 **	1.45 × 10^6^	±27,004 **
Broccoli sprout	unknown	μg/100 g FW	3.11	±0.13 **	7.07 × 10^−2^	±0.00 **	3.92	±0.14 **	8.61 × 10^4^	±3868 **
		μg/100 g DW	56.4	±2.37 **	1.28	±0.03 **	71.1	±2.46 **	1.56 × 10^6^	±70,066 **
Alfalfa bean sprout	unknown	μg/100 g FW	1.70	±0.11 **	1.45 × 10^−1^	±0.01 **	ND		7.82 × 10^4^	±751 **
		μg/100 g DW	32.2	±2.06 **	2.75	±0.14 **			1.48 × 10^6^	±14,210 **
Pea sprout	unknown	μg/100 g FW	1.29	±0.02 **	ND		ND		7.84 × 10^4^	±1812 **
		μg/100 g DW	16.4	±0.19 **					9.98 × 10^5^	±23,062 **
Buckwheat sprout	unknown	μg/100 g FW	4.19	±0.05 **	5.64 × 10^−1^	±0.03 **	4.55 × 10^−1^	±0.03 **	7.41 × 10^4^	±2693 **
		μg/100 g DW	59.4	±0.76 **	8.00	±0.38 **	6.45	±0.36	1.05 × 10^6^	±38,210 **
Apple	Shinano dolce	μg/100 g FW	2.13	±0.06 **	4.13 × 10^−1^	±0.02 **	ND		4.91 × 10^4^	±1905 **
		μg/100 g DW	15.1	±0.45 **	2.92	±0.11 **			3.47 × 10^5^	±13,472
Japanese pear	Twentieth century	μg/100 g FW	2.60	±0.01 **	2.89 × 10^−1^	±0.02 **	ND		2.69 × 10^4^	±719
		μg/100 g DW	19.8	±0.05 **	2.20	±0.13 **			2.05 × 10^5^	±5470
Grape	Nagano purple	μg/100 g FW	3.15	±0.10 **	4.35 × 10^−1^	±0.02 **	ND		6.86 × 10^4^	±186 **
		μg/100 g DW	15.7	±0.50 **	2.17	±0.08 **			3.42 × 10^5^	±928

Ach—acetylcholine; PCh—propionylcholine; BCh—butyrylcholine; Ch—choline; ND—not detected; FW—fresh weight; DW—dry weight; * *p* < 0.05, ** *p* < 0.01 versus eggplant, as evaluated by the Tukey HSD.

**Table 5 foods-09-01029-t005:** Content of choline compounds in 6 cultivars of eggplant (*n* = 3, μg/100 g FW).

Cultivar	ACh		PCh		BCh		Ch	
Higomurasaki	5.53 × 10^3^	±171	7.78	±0.18	2.85	±0.09	3.09 × 10^4^	±720
Chikuyo	5.47 × 10^3^	±88	6.97	±0.15	1.43	±0.04	3.38 × 10^4^	±1419
Senshu mizunasu	5.15 × 10^3^	±102	5.65	±0.04	0.740	±0.04	2.18 × 10^4^	±743
Koryo sarada nasu	4.24 × 10^3^	±107	3.71	±0.07	1.28	±0.05	1.96 × 10^4^	±338
Batten nasu	3.06 × 10^3^	±50	1.53	±0.01	0.604	±0.08	3.63 × 10^4^	±1748
Onaga nasu	2.79 × 10^3^	±112	5.90	±0.10	2.94	±0.04	3.56 × 10^4^	±848

ACh—acetylcholine; PCh—propionylcholine; BCh—butyrylcholine; Ch—choline.

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
