# Peer review of "LC–MS/MS Analysis of Choline Compounds in Japanese-Cultivated Vegetables and Fruits"

_foods, 2020, doi:10.3390/foods9081029_

Round 1

Reviewer 1 Report

This is very well written and interesting paper in which choline compound contents in 19 cultivated fruits and vegetable were identified and determined. The choline compounds were extracted from lyophilized powder samples purified and concentrate using cation exchange column and determined by LC-MS/MS method. The method was validated by determining linearity, limit of detection (LOD), limit of quantification (LOQ), inter- and intraday reproducibility and accuracy. The validated method was used to determine the concentration of choline compounds in 19 types of fresh fruits and vegetables. The obtained results show that all agricultural products contain acetylcholine and choline. While propionylcholine were determined in  17 and butyrylcholine in 12 products. The Authors proved that acetylcholine concentration was 2900-fold higher in eggplants than in agricultural products.

In my opinion the most interesting is the fact that the half-live time of acstylcholine in eggplants was approximately 16 days, ehich is longer than the shelf life of this plant. Therefore they can concluded that eggplants can be a good source of acetylcholine, one of the most important essential nutrient.

Author Response

Thank you very much for your evaluation of our research. For the publication, the format of the table was improved to make the data easy to read As you said, eggplant can play a great role as an effective source of ACh intake in the future. We will also conduct further research on eggplant.

Reviewer 2 Report

The manuscript is poorly written and needs better style redaction. However, the procedure seems interesting and the method used is correct. Therefore, I recommend improvement of the style.

Specific comments:

1) The manuscript needs better redaction;

2) Authors should review the document to change certain expressions

3) More formal style is required

Author Response

I am very grateful for your comments about the manuscript. According to your advice, we amended the relevant part of the manuscript:

  1. Improve the format of the table to make the data easy to read
  2. Page 3 Line 97 and 108: density = 0.990 g/m3 → g/cm3
  3. Table 1: BCh 45, 6 (7.4), CH2→1.45 sext (7.4) CH2
  4. Page 7 Line 202, Page 10 Line 266, Page 13 Line 332: choline compounds → choline esters
  5. Align the significant figures of all tables (3 significant figures)
  6. Added 2.2. General techniques